# Towards layer-selective quantum spin hall channels in weak topological insulator Bi$_4$Br$_2$I$_2$

Jingyuan Zhong[1,6], Ming Yang[1,6], Zhijian Shi[1,6], Yaqi Li[1], Dan Mu[2], Yundan Liu[2], Ningyan Cheng[3], Wenxuan Zhao[4], Weichang Hao [1,5], Jianfeng Wang[1] ✉, Lexian Yang[4] ✉, Jincheng Zhuang [1,5] ✉ & Yi Du[1,5] ✉

Weak topological insulators, constructed by stacking quantum spin Hall insulators with weak interlayer coupling, offer promising quantum electronic applications through topologically non-trivial edge channels. However, the currently available weak topological insulators are stacks of the same quantum spin Hall layer with translational symmetry in the out-of-plane direction—leading to the absence of the channel degree of freedom for edge states. Here, we study a candidate weak topological insulator, Bi$_4$Br$_2$I$_2$, which is alternately stacked by three different quantum spin Hall insulators, each with tunable topologically non-trivial edge states. Our angle-resolved photoemission spectroscopy and first-principles calculations show that an energy gap opens at the crossing points of different Dirac cones correlated with different layers due to the interlayer interaction. This is essential to achieve the tunability of topological edge states as controlled by varying the chemical potential. Our work offers a perspective for the construction of tunable quantized conductance devices for future spintronic applications.

The category of topological quantum materials (TQMs) has been significantly expanded in the past decades. In general, the band evolvements of TQMs are determined by the interplay of various symmetries and microscopic interactions such as spin-orbital coupling (SOC)[1–8]. Among the TQMs, the weak topological insulator (WTI) exhibits unique properties such as even number of topological surface states (TSSs) and directional spin current at selective surfaces[9–13]. Similar to the strong topological insulator (STI), WTI can be described by using the $\mathbb{Z}_2$ topological invariant[9,10]. Distinguished by four $\mathbb{Z}_2$ invariants, the WTI differs from the STI in its surface states nature and anisotropy, which make its surface electronic properties sensitive to the surface orientation. Moreover, the spin current in WTI can be constrained in 1D direction due to the high electronic anisotropy, engendering the ideal prohibition of backscattering and more robust nature against disorder[13,14]. The WTI phase could be formed by stacking quantum spin Hall (QSH) insulators with translational symmetry, in which consists of multiple topological non-trivial edge channels to eliminate the effect of contact resistance in the QSH device applications[14–17]. In principle, the QSH insulators are viewed as the building block to construct different three-dimensional (3D) topological quantum phases with different stacking orders and interlayer coupling strengths, such as STI, WTI, and high-order topological insulator (HOTI)[11,18,19]. Therefore,

[1]School of Physics, Beihang University, Haidian District, Beijing, China. [2]Hunan Key Laboratory of Micro-Nano Energy Materials and Devices, and School of Physics and Optoelectronics, Xiangtan University, Hunan, China. [3]Information Materials and Intelligent Sensing Laboratory of Anhui Province, Key Laboratory of Structure and Functional Regulation of Hybrid Materials of Ministry of Education, Institutes of Physical Science and Information Technology, Anhui University, Hefei, Anhui, China. [4]State Key Laboratory of Low Dimensional Quantum Physics, Department of Physics, Tsinghua University, Beijing, China. [5]Centre of Quantum and Matter Sciences, International Research Institute for Multidisciplinary Science, Beihang University, Beijing, China. [6]These author contributed equally: Jingyuan Zhong, Ming Yang, Zhijian Shi. ✉e-mail: wangjf06@buaa.edu.cn; lxyang@tsinghua.edu.cn; jincheng@buaa.edu.cn; yi_du@buaa.edu.cn

searching for appropriate QSH insulators is an essential for realizing the topological quantum phases and exploring their potential applications in low-consumption electrics and spintronics.

The monolayer $Bi_4X_4$ ($X = Br$, I) has been theoretically predicted and experimentally confirmed to be a large-gap QSH insulator with the energy gap more than 0.2 eV, which is much larger than the thermal activation energy at room temperature[20–23]. Distinctive to other QSH insulators, the monolayer $Bi_4X_4$ possesses the 1D molecule chain elongated along lattice $b$ axis as the structural building block. The weak inter-chain interaction force is also in favor of clean and atomically sharp edges for the preservation of edge states[21,22]. Due to the quasi-1D nature of $Bi_4X_4$, there are two cleavable surfaces, fascinating the measurements of topological properties by detecting the band structures of different surfaces of their 3D allotropes.

Notably, there are three kind of 3D phases in $Bi_4X_4$ system, $\beta$-$Bi_4I_4$, $\alpha$-$Bi_4I_4$, $\alpha'$-$Bi_4Br_4$, constructed by QSH insulators with different stacking orders. For $\beta$-$Bi_4I_4$, only a single $Bi_4I_4$ layer stacks repeatedly along lattice $c$ axis, engendering the degeneracy of the QSH edge states and the consequent formation of WTI phase[13]. For $\alpha$-$Bi_4I_4$ and $\alpha'$-$Bi_4Br_4$, there exist gliding and/or rotation between adjacent layers in the unit cell, resulting in nondegenerate edge states. In this case, the energy gap opens due to the hybridization between the adjacent edge states, leaving metallic hinges states within the gapped surface states and realizing the HOTI state[12,13,20]. These boundary states originate from QSH edge states have evoked various quantum phenomena, such as Tomonaga-Luttinger liquid, anomalous planar Hall effect, and Shubnikov de Haas (SdH) oscillations[24–30]. Thus, the monolayer $Bi_4X_4$ is an ideal unit to expand the topological non-trivial phases by modulating the topology hierarchy.

Here we successfully synthesize $Bi_4Br_2I_2$ crystal with three $Bi_4X_4$ layers in the unit cell, and identify the WTI phase by scanning tunneling microscopy/spectroscopy (STM/STS), angle-resolved photoemission spectroscopy (ARPES), and first-principle calculations. Compared to the previously reported WTI phase constructed by one single block, our triple-layer WTI phase possesses the abundant tunability of the electronic structure due to the interplay between the nondegenerate edge states and interlayer interaction, leading to layer-selective QSH channels that can be directly controlled by modulating the chemical potential. Our work inspires the exploration of the topological phases by stacking QSH blocks, and offer the path to realize the proof-of concept devices that require the channel degree of freedom based on helical edge states.

## Results

### Structure of $Bi_4Br_2I_2$

Figure 1a shows the X-ray diffraction (XRD) spectra of two cleavable planes of $Bi_4Br_2I_2$ crystal with sharp diffraction peaks, confirming the crystal structure of our sample. The insets of the Fig. 1a show the optical images of (001) surface and (100) surface, where the smoothness of (001) surface is clearly better than (100) surface, which is due to the larger intra-layer (along $a$ axis) binding energy than inter-layer (along $c$ axis) binding energy[23]. The mole ratio of different elements is obtained by energy dispersive spectroscopy (EDS) measurements (See Supplementary Note 1). The large-area STM image of the (001) surface in Fig. 1b shows the large area of clean surface after cleavage that is ideal for ARPES measurements. In the high-resolution STM image (Fig. 1c), there are two kinds of atoms (dark and bright) distributing randomly on the (001) surface, which is due to the different radius of Br and I ions (See Supplementary Note 2 for details). Figure 1d shows the high-angle annular dark-field scanning transmission electron microscopy (HAADF-STEM) measurements on the uncleavable (010) plane, where the triple-layered structure can be clearly identified. We define these three layers as $A_1$, $A_2$, and B, respectively, as displayed in Fig. 1d. The $A_1$-$A_2$ stacking is the same as that of the adjacent layers in $\alpha$-$Bi_4I_4$ with $b/2$ shift with respect to each other, while the $A_2$-B stacking resembles the interlayer arrangement in $\alpha'$-$Bi_4Br_4$, as schematically shown in the schematics in Fig. 1e and the HAADF-STEM images in Supplementary Note 3. All the structural characterizations indicate a new stacking order combining the building sequences of $\alpha$-$Bi_4I_4$ and $\alpha'$-$Bi_4Br_4$ in $Bi_4Br_2I_2$ crystal.

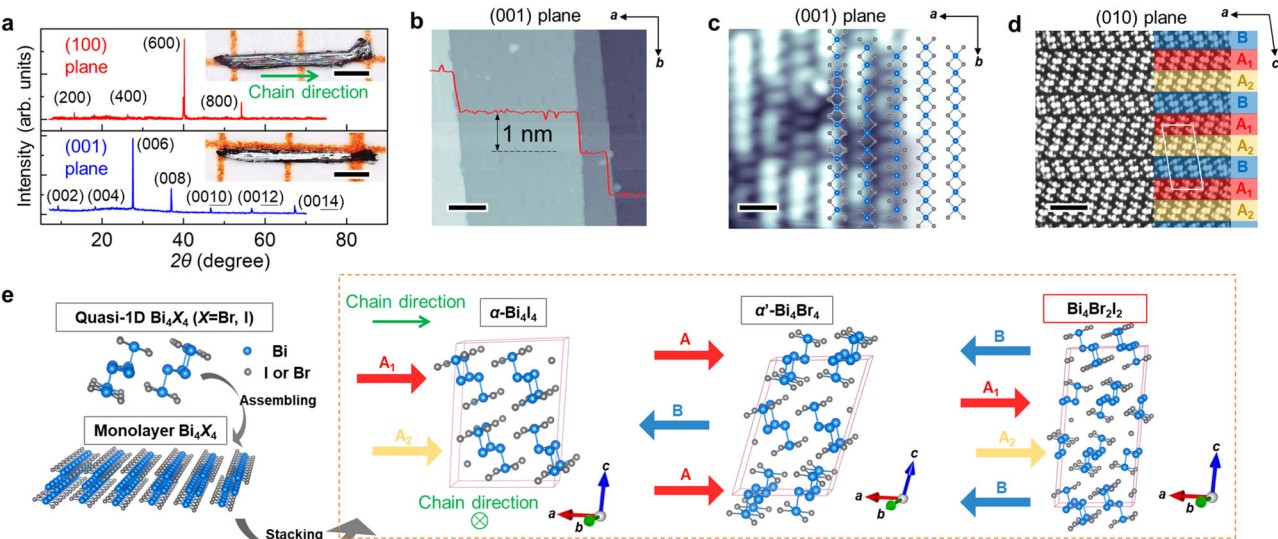

**Fig. 1 | Crystal structure of $Bi_4Br_2I_2$. a** XRD spectra of (100) plane (upper panel) and (001) plane (lower panel), respectively. The insets show the optical images of two cleaved surface with the green arrow indicating the chain direction. Scale bars are 0.5 mm. **b** Large-area STM topography of cleaved (001) plane with the step height ~1 nm. The scale bar is 12 nm. **c** High-resolution STM image of (001) plane with the projected schematic of atom model. The scale bar is 1 nm. **d** HAADF-STEM results of (010) plane with alternating $A_1$, $A_2$, and B layers. The white rhomboid represents the triple-layer unit cell. The scale bar is 1.7 nm. **e** (001) monolayer of $Bi_4X_4$ topological materials assembled by quasi-1D molecular chain as the building block along $a$ axis by vdW force. The blue and gray balls represent Bi atoms and I/Br atoms, respectively. The schematics of the stacking mode along $c$ axis and crystal structure of $\alpha$-$Bi_4I_4$, $\alpha'$-$Bi_4I_4$, and $Bi_4Br_2I_2$ are shown in the orange dashed square. Each of the red, yellow, and blue parallel arrow represents one (001) plane of $Bi_4X_4$ with the orientations and displacements.

## QSH nature of monolayer Bi₄Br₂I₂

Before the experimental investigation, we study the topological properties of $Bi_4Br_2I_2$ using ab-initio calculation. Our calculations of the electronic structure of $Bi_4Br_2I_2$ nanoribbon and $\mathbb{Z}_2$ topological invariant reveal band inversion with parity change driven by SOC, demonstrating the QSH nature of monolayer $Bi_4Br_2I_2$ (See Supplementary Note 4). Figure 2a shows the bulk and surface projected Brillouin zone (BZ) labeled with time-reversal invariant momenta (TRIM) points of bulk $Bi_4Br_2I_2$. The density functional theory (DFT) calculations without SOC in Fig. 2b implies the semiconducting nature of bulk $Bi_4Br_2I_2$. The calculated results with the full high-symmetry lines could be identified in Supplementary Note 5. There are three groups of bulk valence bands (BVBs) and bulk conduction bands (BCBs) originate from three different $Bi_4X_4$ layers in one unit cell. After turning on the SOC, the parities of BVBs change from $(- - +)$ to $(+ - +)$ and $(- + -)$ to $(+ - +)$ at $M$ and $L$ points, respectively. The bandgap slightly shrinks and the band inversion of the constituents with opposite parities of BVBs and BCBs occurs for three times at both $M$ point and $L$ point of bulk BZ, which is identical to the one-time band inversion at these two TRIMs and indicates the WTI nature of bulk $Bi_4Br_2I_2$[9,10]. The WTI nature is further confirmed by the calculations of $\mathbb{Z}_2$ topological invariant and surface states using Wannier function (See Supplementary Note 5).

We perform laser-ARPES measurements to experimentally investigate the electronic structures of the in-situ cleaved $Bi_4Br_2I_2$ crystal. Figure 2c shows the constant energy contours (CEC) of (001) plane measured at 77 K with different binding energies. We observe the highly anisotropic features extending along $\bar{\Gamma} - \bar{M}$ direction, similar to the previous reports of band structures of $Bi_4X_4$ and consistent with the quasi-1D lattice structure of this system[12–20]. The energy-

momentum dispersions along $k_y$ around $\bar{\Gamma}$ and $\bar{M}$ are displayed in Fig. 2d, e, respectively. The band top of BVB locates at about 0.4 eV and 0.1 eV at the $\bar{\Gamma}$ and $\bar{M}$ points, respectively. We also observe the bottom edge of the BCB near the Fermi surface at $\bar{M}$ point (Fig. 2e), where an approximate 0.1 eV gap to BVB is identified by the energy distribution curve (EDC) spectra in Fig. 2f. The ARPES measurement with Helium light source ($h\nu$ ~ 21.2 eV) at 7 K is also measured to confirm the energy gap (Supplementary Note 6), where the BCBs and 0.1 eV gap at $\bar{M}$ point could be clearly distinguished due to the downward shift of the overall band structure correlated to the Lifshitz transition effect in this system[22]. The semiconducting nature is further confirmed by STS spectrum measured at the terrace of the cleaved (001) surface, as displayed in Fig. 2g. Moreover, compared to the gap feature in the terrace, the STS spectrum detected at the edge position shows an obvious nonzero density of state (DOS) filling in the gap, implying the presence of the edge state and consistent with the QSH insulating nature of monolayer $Bi_4Br_2I_2$.

## Coupled TSS of (100) surface

The 3D plot of ARPES spectra on (100) surface in ($k_y$, $k_z$, $E$) space is shown in Fig. 3a to further investigate the topological properties of $Bi_4Br_2I_2$. The linear-like band dispersion can be identified in the CEC at the different binding energies in Fig. 3b, implying a larger anisotropy of band structure of (100) surface due to the weaker interchain interaction compared to (001) surface[27,28,31]. Moreover, there are more than one group of linear-like band observed in CEC results, indicating the abundant band structure of (100) surface. The energy-momentum dispersions along the chain direction at TRIM of (100) surface BZ, $\bar{\Gamma}$ and $\bar{Z}$, exhibit the same linear tendency as shown in Fig. 3c–f. This consistently linear behavior implies the topological surface states at

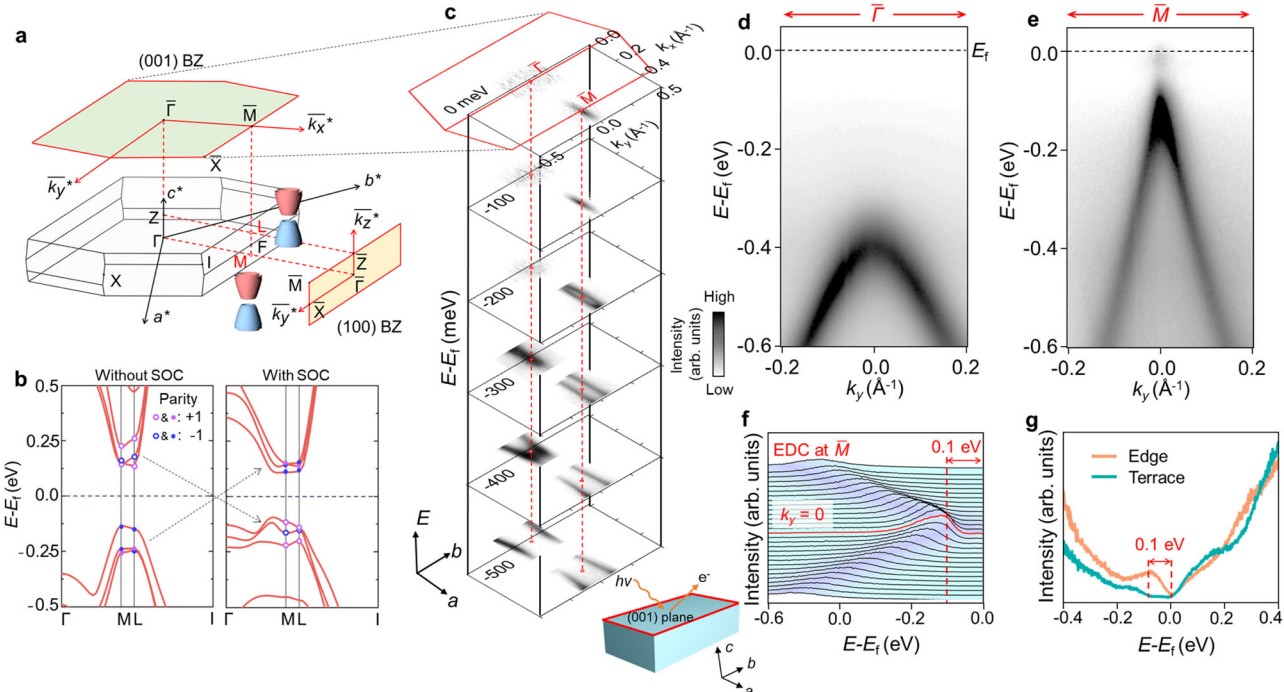

**Fig. 2 | Semiconducting nature of $Bi_4Br_2I_2$. a** Bulk BZ of $Bi_4Br_2I_2$ with time-reversal symmetry invariant momenta (TRIM). The red hexagon and square represent the projected (001) BZ and (100) BZ, respectively. The schematics label the band inversion TRIMs $L$ and $M$. **b** The calculated bulk band structure of $Bi_4Br_2I_2$ (left panel) without spin-orbital coupling (SOC) and (right panel) with SOC. The purple and blue circles label the even (+) parities and the odd (−) parities, respectively. The dashed arrows are marked for the indication of band inversion by SOC. **c** Constant energy contour (CEC) of (001) surface at different binding energies. The inset

shows the experimental setup of the measurements of the (001) plane. The color bar applies for **c**–**e**. **d**, **e** Energy-momentum dispersion along $k_y$ direction at and, respectively. The black dashed lines indicate the Fermi surface. **f** Energy distribution curve (EDC) spectra with the red dashed line denoting the gap size to Fermi surface which is about 0.1 eV. **g** STS spectra of (001) surface collected at terrace (green line) and edge (orange line). The red arrow and dashed lines are applied to denote the 0.1 eV gap.

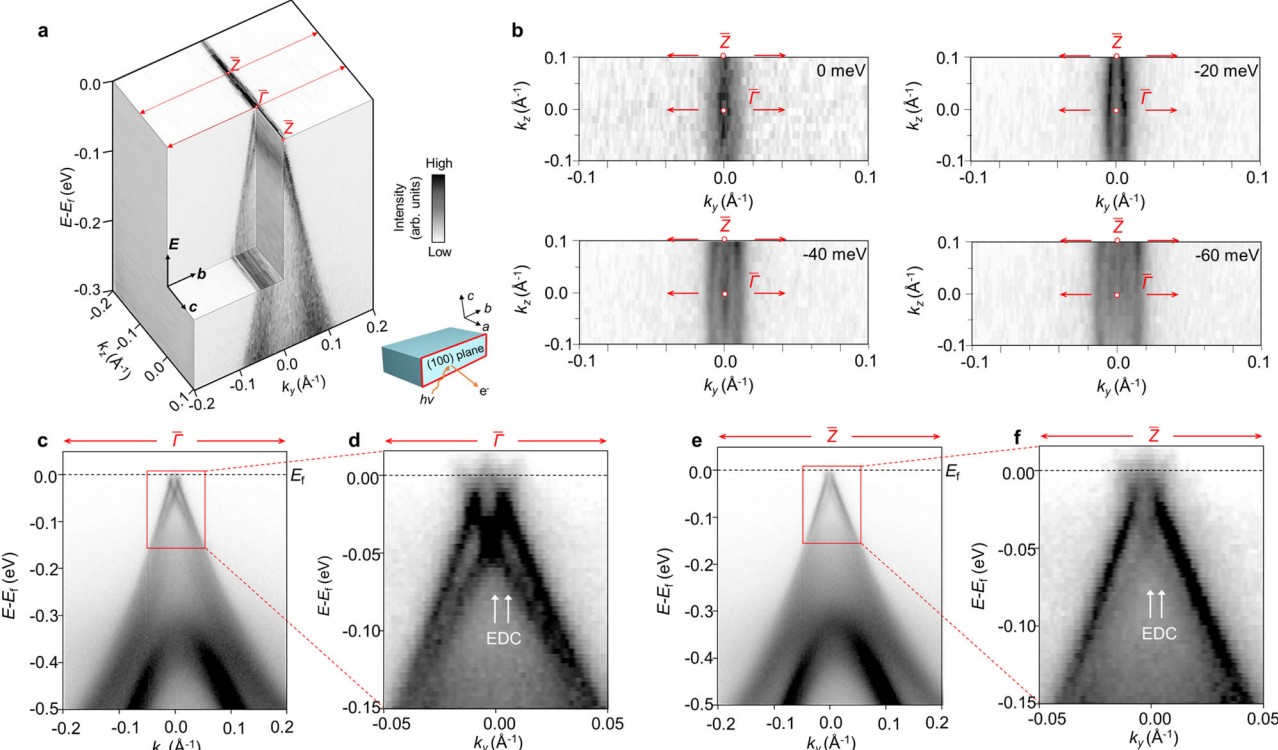

**Fig. 3 | ARPES results of (100) surface. a** 3D illustration of band structure of (100) surface with the projected TRIM and. The inset shows experimental setup of the measurement of (100) surface. The color bar applies for **a**–**f**. **b** Constant energy contours (CECs) at the binding energies of 0 meV, 20 meV, 40 meV, and 60 meV. The red arrows indicate the energy-momentum dispersion along $k_y$ at and (labeled by red circles). **c** Energy-momentum dispersion along $k_y$ direction at and **d** enlarged spectra. **e** Energy-momentum dispersion along $k_y$ direction at and **f**, enlarged spectra. The white arrows indicate the location of characteristic energy distribution curves (EDCs).

(100) surface are highly conducting along the chain direction. In order to obtain more information on (100) band structure, we perform the fitting analysis of the EDCs at different momenta, as shown in Fig. 4a, c (more fitting details see Supplementary Note 7). The EDC at $\bar{\Gamma}$ and $\bar{Z}$ ($k_y = 0$) are decomposed to three Lorentzians corresponding to the BVB and two Dirac points, respectively. Distinguishingly, the EDC slightly off $\bar{\Gamma}$ with $k_y = 0.005$ Å$^{-1}$ and off $\bar{Z}$ with $k_y = 0.007$ Å$^{-1}$ are divided into five Lorentzians. In this case, the signal from the upper crossing point is made up by three peaks, where the two extra peaks flank at the binding energy around 18 meV for spectra in both Fig. 4a, c. The extracted peak positions at different momenta are labeled on the EDC at $\bar{\Gamma}$ and $\bar{Z}$, as shown in Fig. 4b, d, respectively. Three branches of linear band dispersion could be identified, which agrees well with our calculation results of triple band inversion at both $\bar{\Gamma}$ and $\bar{Z}$. The fitting results demonstrate the gapless feature of the Dirac point at TRIM protected by time-reversal symmetry and inversion symmetry. The two extra fitting peaks flanking around 18 meV are the reflection of the opened gap originating from the weak interlayer coupling at the crossing point.

## Layer-selective QSH channels

The DFT calculations are applied to figure out the contributions of different layers on these three groups of the linear band dispersion. Figure 5a, c display the calculated results of the surface states of different layers along $\bar{\Gamma} - \bar{X}$ and $\bar{Z} - \bar{M}$ directions, respectively, where the red, yellow, and blue symbols with the size proportional to the spectral weight represent the projection from A$_1$, A$_2$, and B layers, respectively. Both of the calculated results in Fig. 5a, c show that there are three pairs of homochromatic linear bands with three gapless Dirac points from A$_1$, A$_2$, and B layers approximately located at around −40 meV, 18 meV, and 0 meV, respectively. The band hybridization in Fig. 5a

could be indicated by the formation of band gap labeled by the purple arrows at the crossing points of every two of the three topological surface states, resulting from the interlayer interactions. The enlarged band structures with the comparison with ARPES results are exhibited in Fig. 5b, d, where the high consistency confirms the triple-layer WTI nature of Bi$_4$Br$_2$I$_2$ (additional calculated results can be seen in Supplementary Note 8). For the band dispersion along $\Gamma - X$ direction, the gap size at the band crossing points corresponding to A$_1$ and A$_2$ layers, A$_1$ and B layers, and A$_2$ and B layers are ~0.5 meV, 1 meV, and 10 meV, respectively, implying the largest coupling strength between A$_2$ and B layers. It should be noted that the calculated gap value of 10 meV is close to the calculated gap value of second-order topological insulator Bi$_4$Br$_4$[28], which is reasonable after considering the fact that the stacking order of Bi$_4$Br$_4$ is A$_2$B stacking, as shown in Fig. 1e.

WTI requires the odd number of Dirac bands at arbitrary binding energy residing in the gap region. Therefore, we securitize on the numbers of Dirac bands at three typical binding energies, 48 meV, 10 meV, and −19 meV, labeled by $E_1$, $E_2$, and $E_3$, respectively, where $E_2$ and $E_3$ reside in the energy gap region of two gaps induced by A$_2$B hybridization and A$_1$B hybridization, respectively. The number values of Dirac bands are three, one, and one, respectively, confirming the WTI nature of Bi$_4$Br$_2$I$_2$. The real space distribution of the topological non-trivial surface states is also studied at these three energy levels, as displayed in Fig. 5e–g. The surface state distributes uniformly in the entire (100) surface at $E_1$, as shown in Fig. 5e, which is identical to WTI case of β-Bi$_4$I$_4$ constituted by one layer under translational symmetry. The penetration length of surface state along lattice $a$ axis is around twice width of the chain, which is consistent with the width of 1D edge state of monolayer Bi$_4$Br$_4$[21,22] and consequently demonstrates that the (100) surface state originates from the stacking of edge states of monolayer Bi$_4$Br$_2$I$_2$. Therefore, all three layers contribute one pair of

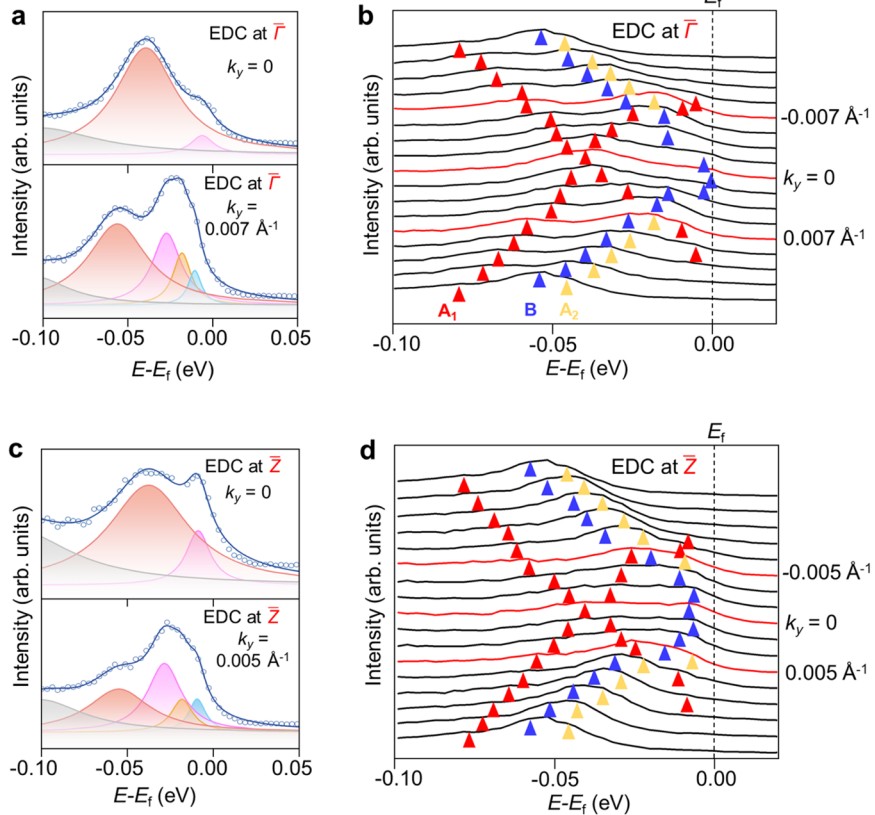

**Fig. 4 | Coupling of TSS. a** Characteristic energy distribution curves (EDCs) at $k_y = 0$, and 0.007 Å$^{-1}$ (blue circles) with Lorentzian fitting results (solid lines) and **b** EDC stacking spectra at. **c** Characteristic EDCs at $k_y = 0$, and 0.005 Å$^{-1}$ with Lorentzian fitting results and **d**, EDC stacking spectra at. The red, yellow, and blue triangles are marked for the locations of fitted peaks, representing the topological surface states of A$_1$, A$_2$, and B layers, respectively.

spin-momentum-locked conducting channels, as shown in the lower panel of Fig. 5e. Interestingly, the conducting channels of A$_2$ and B layers would be turned off at $E_2$ due to the gap induced by the coupling of these two layers, leaving only the QSH channels carried by A$_1$ layer, as shown in Fig. 5f. $E_3$ locates at the gap evoked by the hybridization between A$_1$ and B layers, where only the QSH channels derived from A$_2$ layers are switched on. More distributions of the QSH channels as a function of binding energy could be founded in Supplementary Note 9. It should be noted that Fig. 5e–g is the (total) charge distributions of topological surface states at one point (or three points) in the BZ, which are the crossing points between three surface bands and the selected binding energy along $\Gamma - X$ direction. More accurate calculations of charge distributions based on a very small energy range around the selected binding energies in the entire half BZ are displayed in Supplementary Fig. 15 in Supplementary Note 9, which show almost the same results as in Fig. 5e, f, and consequently conform the validity of our results. Since the energy differences between different gaps are relatively small (several tens meV), it is feasible to tune the Fermi surface at different gap regions simply by electric gating method or charge carriers doping, as shown in the schematic in Fig. 5h. Consequently, the non-degeneracy of the three channels and the interlayer interaction-induced the energy gap provide the additional degree of freedom to control the QSH channels in selective layers in Bi$_4$Br$_2$I$_2$.

## Discussion

We now discuss the mechanism of non-degeneracy of the three QSH channels. The non-degenerate QSH channels are correlated to the non-degenerate surface states of (100) plane derived from the bulk band inversions of three different layers, A$_1$, A$_2$, and B. The translational symmetry along $c$ axis protects the degenerating features of bulk

bands. The translational symmetry is changed from monolayer type in $\beta$-Bi$_4$I$_4$ to triple-layer type in Bi$_4$Br$_2$I$_2$. As a consequence, the crystal field faced by each layer is expected to be different due to the degradation of translational symmetry, engendering the non-degeneracy of these three layers. Furthermore, this non-degeneracy may be enhanced in the condition if the chemical composition varies at different layers and the strain effect works. The EDS mapping results (Supplementary Note 3) imply that more I content is identified in the region of the lower surface of A$_1$ layer and the upper surface of A$_2$ layer. This phenomenon conforms to the A$_1$A$_2$ stacking structure of pure Bi$_4$I$_4$ at low temperature[12].

By introducing the interlayer coupling to the edge state model of a QSH (Supplementary Note 10), here we build a surface state Hamiltonian to reveal the physical process accounting for the (100) surface states of Bi$_4$Br$_2$I$_2$, which can be written as:

$$H_S = \begin{pmatrix} \hbar v_{F1} k_y + \varepsilon_1 & 0 & 0 & -m_{12} & 0 & -m_{13} \\ 0 & -\hbar v_{F1} k_y + \varepsilon_1 & -m_{12} & 0 & -m_{13} & 0 \\ 0 & -m_{12} & \hbar v_{F2} k_y + \varepsilon_2 & 0 & 0 & -m_{23} \\ -m_{12} & 0 & 0 & -\hbar v_{F2} k_y + \varepsilon_2 & -m_{23} & 0 \\ 0 & -m_{13} & 0 & -m_{23} & \hbar v_{F3} k_y + \varepsilon_3 & 0 \\ -m_{13} & 0 & -m_{23} & 0 & 0 & -\hbar v_{F3} k_y + \varepsilon_3 \end{pmatrix}$$

$$(1)$$

where $m$ is the coupling between arbitrary two different layers (denoted by subscripts and 1, 2, and 3 represent the A$_2$, B, and A$_1$ layers, respectively), and $\varepsilon$ and $v_F$ are the onsite potential and Fermi velocity of different layers, respectively. Three helical edge states with interlayer-coupling gaps are formed on the (100) surface of Bi$_4$Br$_2$I$_2$, which is roughly consistent with our DFT calculations (Supplementary Note 10). We fit our model to the DFT calculated band structure to

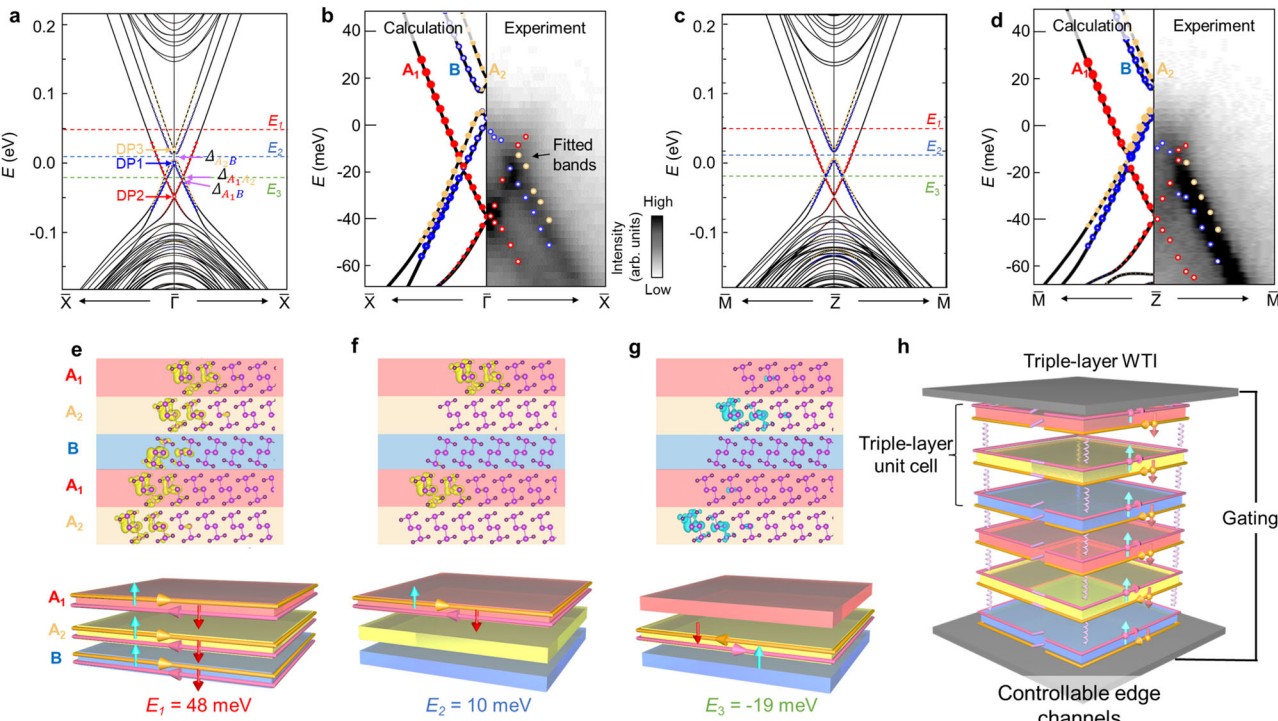

**Fig. 5 | Layer-selective QSH channels. a** DFT calculated band structure of (100) surface along the direction. The band weights of $A_1$, $A_2$, and B layers are denoted as red, yellow, and blue circles, respectively. The three Dirac points (DPs) are labeled by arrows with corresponding color of different layers. The red, blue, and green dashed lines label three binding energy locations of 48 meV, 10 meV, and −19 meV, respectively, which also apply to **c**, **e**–**g** The positions of three gaps induced by interlayer coupling are indicated by purple arrows. **b** Enlarged view of the calculated bands (left panel) in **a** and corresponding ARPES spectra with the fitted results (right panel). The color bar applies for **b**, **d**. **c**, **d** Calculated topological surface

states along direction and its enlarged picture with experimental results. **e**–**g** Top panel: the charge distribution of topological surface states at different binding energies labeled by the dashed lines in **a** and **c**, where the yellow and blue are marked for the opposite direction of group velocity at positive $k_y$. Lower panel: schematics of layer-selective QSH channels at different binding energies. The pink and orange arrows indicate the direction of current, the blue and red arrows represent the spin orientations. **h** Schematic of proof-of-concept device with gating controllable layer-selective QSH channels based on triple-layer WTI. The purple springs represent the interlayer interaction.

obtain the reasonable parameters, as listed in the caption of Supplementary Fig. 16. All the physical parameters, $m$, $\varepsilon$ and $v_F$, are different for three layers, confirming the non-degeneracy of three elemental layers form the view of the physical model. Furthermore, our model could build a unified understanding of the formation of (100) surface states for $Bi_4X_4$ family of materials (Supplementary Note 10).

Apart from the previously reported WTI stacked by the degenerate QSH layer, $\beta$-$Bi_4I_4$, the $Bi_4Br_2I_2$ with triple-layer structure possesses unique electronic properties for both fundamental research and potential electric-device applications. The triple-times inversions at two TRIMs of BZ of bulk band structure bestow the degree of tunability on the topological properties of this system by external perturbations. For example, the strain effect has been certified to be an effective means to realize the topological phase transition by controlling the bulk band inversion[11]. The multiple band inversions make $Bi_4Br_2I_2$ a fertile ground to realize the various topological phases, such as HOTI, STI, and dual topological phases, after applying the strain effect to tune the band inversions. The controllable QSH channels in the thin flake of $Bi_4Br_2I_2$ offer the possibility to realize the quantized conductance with multiple values and special distributions in nanoscale size, which is a potentially useful platform for the semiconductor industry. Furthermore, the quasi-1D nature can enhance the electron-electron interaction strength, which could result in the helical Luttinger-liquid (LL) behavior in the edge states of monolayer $Bi_4Br_2I_2$ and the consequent coupled 2D LL behavior in the (100) surface states with weak interlayer interaction.

The triple-layer WTI $Bi_4Br_2I_2$ possesses the unique three pairs of topological states coupled with each other, resulting in three gapped

bands with different gap sizes due to the interlayer coupling. The coupling behavior and non-degeneracy in triple-layer $Bi_4X_4$ topological materials provide encouraging perspectives to control the degree of channel freedom derived from the 1D topological edge states. Our work paves the way to realize layer-selective QSH channels in self-assembling vdW materials by tuning the Fermi level, which may inspire further dedicated transport measurement to achieve adjustable quantized conductance.

## Methods

### Synthesis of high-quality monocrystal $Bi_4Br_2I_2$
Solid-state reaction was applied to grow $Bi_4Br_2I_2$ monocrystal. Highly pure Bi, $BiI_3$, and $BiBr_3$ (the mole ratio of $BiI_3$ and $BiBr_3$ are 1:1) powders were mixed under Ar atmosphere in a glovebox and sealed in a quartz tube under vacuum. The mixture was placed in a two-heating-zone furnace with a temperature gradient from 558 K to 461 K for 72 h. Monocrystal $Bi_4Br_2I_2$ nucleated at the high-temperature side of the quartz after cooling down.

### XRD characterization
XRD measurements were performed in AERIS PANalytical X-ray diffractometer at room temperature. The cleaved (100) or (001) surfaces of $Bi_4Br_2I_2$ were carefully set to parallel with the sample stage, and the divergence slit is 1/8° with beta-filter Ni to remove the $K_\beta$ wavelength of Cu radiation.

### HAADF-STEM characterization
For STEM characterization, first, a TEM lamella was prepared by using a Zeiss Crossbeam 550 FIB-SEM. Then, the characterization was

conducted on a probe and image-corrected FEI Titan Themis Z microscope equipped with a hot-field emission gun working at 300 kV.

## STM characterization

The STM measurements were carried out using a low-temperature UHV STM/scanning near-field optical microscopy system (SNOM1400, Unisoku Co.), where the bias voltages were applied to the substrate. The differential conductance, $dI/dV$, spectra were acquired by using a standard lock-in technique with modulation at 973 Hz. All the STM measurements were performed at 77 K.

## Laser-based ARPES

Laser-based ARPES measurements were performed using DA30L analyzers and vacuum ultraviolet 7 eV lasers at 77 K in Tsinghua University. The overall energy and angle resolutions were set to 3 meV and 0.2°, respectively. The samples were cleaved in situ and measured under ultra-high vacuum below $6.0 \times 10^{-11}$ mbar.

## ARPES characterization with Helium light

ARPES measurement is performed on fresh in-situ cleaved thick and homogenous crystal, stuck by torr seal glue with exposed (001) or (100) surface on the sample holder. The Helium light ARPES characterizations were performed at $T = 6$ K at the Photoelectron Spectroscopy Station in the Beijing Synchrotron Radiation Facility using a SCIENTA R4000 analyzer with photon energy ~ 21.2 eV. The total energy resolution was better than 15 meV, and the angular resolution was set to ~0.3°, which gave a momentum resolution of ~0.01 π/a. The laser ARPES measurement is conducted in Tsinghua

## First-principles calculations

The first-principles calculations are performed using the Vienna ab initio simulation package[32] within the projector augmented wave method[33] and the generalized gradient approximation of the Perdew-Burke-Ernzerhof[34] exchange-correlation functional. The energy cutoff of 300 eV is used, and $9 \times 9 \times 1$ and $13 \times 13 \times 2$ $\Gamma$-centered $k$-grid meshes are adopted for structural relaxation and electronic structure calculations, respectively. Employing the experimental lattice constants, the crystal structure of $Bi_4I_2Br_2$ is relaxed with van der Waals correction until the residual forces on each atom are less than 0.001 eV/Å. The virtual crystal approximation[35] is employed with the weight of 0.5 for both the Br and I element in all Br/I sites. The SOC effect is considered in our calculations. A tight-binding (TB) Hamiltonian based on the maximally localized Wannier functions (MLWF)[36] is constructed to further calculate the surface states, CEC, and WCC using the WannierTools package[37].

**Uncertainty of Lorentzian peak fitting.** Each fitted peak position contains energy and momentum uncertainty. The uncertainty of momentum mainly comes from angle resolutions (0.2°) of laser ARPES measurement, which is approximately 0.003 Å$^{-1}$. The energy uncertainty from Lorentzian fitting (Lorentzian peak position) is ~ 4 meV, which is close to the resolution of laser ARPES measurement with ~ 3 meV.

## Data availability

Source data needed to evaluate the conclusions are provided in this paper. Additional data related to this paper are available from the corresponding author upon request.

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

## Acknowledgements

This work was supported by the National Natural Science Foundation of China (11904015, 12004030, 12074021, 12274016, 52073006, 12274251), National Key R&D Program of China (2018YFE0202700), the Fundamental Research Funds for the Central Universities (Grant No. YWF-22-K-101).

## Author contributions

J. Zhuang and Y.D. planned the experimental project. J. Zhong, M.Y., and W.Z. conducted ARPES experiments and analyzed the data. D.M. Y. Liu, and Y. Li performed STM/STS experiments and analyzed the data. J. Zhong and M.Y. made and characterized single crystals. N.C. performed the HAADF-STEM experiments. Z.S., W.H., and J.W. conducted the DFT calculations and model analyses. J. Zhong, M.Y., Z.S., L.Y., Y.D., and J. Zhuang wrote the paper. J. Zhuang supervised this work. All authors discussed the results and commented on the manuscript.

## Competing interests

The authors declare no competing interests.

## Additional information

**Peer review information** *Nature Communications* thanks Peng-Fei Liu, and the other, anonymous, reviewer(s) for their contribution to the peer reviews of this work. A peer review file is available.

