## [Peer Review File · Nature Communications]

Reviewers' Comments:

Reviewer #1:

Remarks to the Author:

In the manuscript, the authors explore a new weak topological insulator, trilayer bismuth halide $\text{Bi}_4\text{Br}_2\text{I}_2$, which is stacked by three different QSH insulators. Based on the angle-resolved photoemission spectroscopy and first-principle calculations, they show that the interlayer interaction can induce the band gap opening at the Dirac points and accordingly drive the on/off status of topological edge states. They give a new way to control the quantized conductance for spintronics applications. The work sounds interesting and academic. However, there are several concerns they need to clarify.

1. Certainly, there are 4 invariants distinguishing 16 phases with two general classes: weak (WTI) and strong (STI) topological insulators in three dimensions. The authors should highlight the significance of WTI in contrast to STI.
2. Some additional explanations are needed concerning the calculation conditions. The use of low accuracy energy cutoff of 300 eV must be justified. What is the size of k-point meshes? I see they use the tight-binding (TB) Hamiltonian based on MLWF and the corresponding band structures should be given for the comparison with first-principles.
3. The calculated band structure of $\text{Bi}_4\text{Br}_2\text{I}_2$ with and without SOC should be given in the supporting information along the suggested high-symmetry lines [Comp. Mat. Sci. 128, 140 (2017) or Comp. Mat. Sci. 49 299–312 (2010)].
4. I am confused concerning Fig. 4h. Do the authors mean electric gating method or charge carriers doping can tune the Fermi surface at different gap regions? If so, please give the corresponding first-principles Fermi surfaces at charge carriers doping or electric gating method.
5. Are Fig.4e-4g the charge distribution of topological surface states at one point in the Brillouin zone for one selected binding energy? More details should be given in the manuscript. I remain puzzled by the connection between Fig.4e-4g and Fig. 4h.
6. In the manuscript, the authors explore $\text{Bi}_4\text{Br}_2\text{I}_2$ as a WTI in the light of theory and experiments. It in current form looks like a laboratory report. The authors should highlight the findings and Influence in this field.
7. Besides, I would like the authors illustrate their conclusion from the physical model.

Reviewer #2:

Remarks to the Author:

This manuscript reports on a multi-technique (XRD, STM, STEM, ARPES) characterization a Bi_4X_4 -variant heterostructure based on $\text{Bi}_4\text{Br}_2\text{I}_2$ monolayers with a unique triple-layer stacking arrangement. While the ARPES of the (001) cleave-surface of the Gamma and M-bar points look very similar to the half-dozen other Bi_4I_4 and Bi_4Br_4 ARPES measurements dating back to 2015, what stands out in this work is the exceptional clarity of the ARPES spectra for the side-cleave (100) surface that reveals a cluster of three linear Dirac crossing points just below the Fermi level, which generically looks like two Dirac cones shifted in energy relative to each other.

The authors are able to reproduce the complex (100) ARPES spectra with DFT band theory, including the prediction of a third small-energy-shifted linear band dispersion, which is also experimentally inferred from the ARPES EDC lineshape with Lorentzian peak-fitting. The DFT calculations then go beyond the experimental resolution to elucidate the predicted tiny energy gappings versus degeneracies at the various linear band crossing points. DFT real-space projections of the charge densities at the different Dirac energies finds a clear localization topological surface states to specific layers of the triple-layer structure. This then leads to the logical conclusion that by tuning the Fermi-level higher or lower in energy, e.g. by gating, one could select which layer's topological edge-state contributes to the Fermi-level conduction.

The presented ARPES data are of high quality, and the auxiliary STEM and STM data are very good also. The theory presentation is clear with associated schematic figures. Hence I can recommend publication of the current manuscript as is.

I find that another ARPES manuscript studying the same Bi₄Br₂I₂ material and trilayer stacking is also posted on the arXiv (Noguchi et al., arXiv:2301.07158). Again, their (001) surface results making use of laser and synchrotron energies are very similar to the other binary Bi₄X₄ ARPES studies and give the same energetics of the Gamma and M-bar bulk valence band maxima as in this study. They do not observe the detailed trilayer-splitting of topological surface state bands on the side-cleave (100) surface. This is perhaps related to their observation of both (001) and (100) surfaces being exposed after cleavage and having to use 1 micron beamspot nano-ARPES to help select which top plane is being measured. Sounds like a materials synthesis issue.

Minor: "linear" is mistyped as "liner" in a few places.

Reviewer #3:

Remarks to the Author:

Zhong et al. present structural and spectroscopic data, as well as electronic structure calculations, for the novel topological quantum material Bi₄Br₂I₂, built from the stacking of large-gap quantum spin Hall (QSH) BiX₄ layers. The stacking of the layers in Bi₄Br₂I₂ is unlike that of the previously known BiI₄ and BiBr₄ members of the same family. The authors argue that Bi₄Br₂I₂ is a weak topological insulator, and that the topological edge states of the three distinct BiX₄ layers in the unit cell hybridize, giving rise to a unique gap structure. This conclusion is based on high-resolution ARPES data and on first-principles calculations.

The crucial point here is the description of the surface states of the (100) surface. I find the interpretation of the ARPES results reasonable and qualitatively consistent with the experimental data. The quantitative conclusions rest on the detailed line shape analysis of the spectra of Fig. 3 (h,j). However, considering the level of accuracy required by the very small predicted gaps, a 5-peak fit provides at most circumstantial evidence. In the end, one really should trust the electronic structure calculations, and take a leap of faith (which, incidentally, is not uncommon for similar published ARPES work).

That said, the growth of high quality crystals of a novel topological material is an achievement. Moreover, the suggestion that the Fermi level could be tuned into non-equivalent regions of the surface electronic structure, opens interesting perspectives for the control of the spin conductance of the one-dimensional topological states, and should stimulate dedicated transport measurements. Therefore, I think that the paper could be published in Nature Commun. after the authors have addressed the following points.

- Given the importance of the electronic structure calculations, the authors should discuss how they took into account the random Br/I substitution within the layers. In particular, are they confident that the associated random potential does not smear or otherwise modify the surface gap structure?

Minor points:

- The description (lines 114-115) of Fig. 2(f) is poorly phrased, and as such inaccurate

- The A1-A2-B labelling of the layers in Fig. 1(d) and 1(e) is inconsistent (twice as many labels in (e))

- In Supplementary Note 6, "EF moves downward" should be "bands move downward" or else "EF moves upward"

Response to reviewers' comments

Reviewer 1:

In the manuscript, the authors explore a new weak topological insulator, trilayer bismuth halide $\text{Bi}_4\text{Br}_2\text{I}_2$, which is stacked by three different QSH insulators. Based on the angle-resolved photoemission spectroscopy and first-principle calculations, they show that the interlayer interaction can induce the band gap opening at the Dirac points and accordingly drive the on/off status of topological edge states. They give a new way to control the quantized conductance for spintronics applications. The work sounds interesting and academic. However, there are several concerns they need to clarify.

Comment 1: Certainly, there are 4 invariants distinguishing 16 phases with two general classes: weak (WTI) and strong (STI) topological insulators in three dimensions. The authors should highlight the significance of WTI in contrast to STI.

Reply 1: Thanks for reviewer's appreciation and valuable suggestions on our work. We agree well with the reviewer's opinion that the significance of WTI compared to STI should be described in the manuscript. In three dimensions, topological insulators (TIs) are classified by four Z_2 invariants ($\nu_0; \nu_1, \nu_2, \nu_3$). A strong TI (STI), characterized by nontrivial ν_0 , has metallic surface states with an odd number of Dirac cones in the whole Brillouin zone. The surface states are robust to perturbations that do not break the time-reversal (TR) symmetry. A weak TI (WTI), characterized by $\nu_0 = 0$ but $\nu_1 + \nu_2 + \nu_3 \neq 0$, is topologically equivalent to a stack of 2D TI layers and has an even number of Dirac cones on the side surface. Although its surface states were initially assumed to be unstable with respect to disorders¹, the following research²⁻³ have revealed that surface states of a WTI behave robustly even under strong TR-invariant disorders.

In contrast to STI, the WTI shows unique sensitivity of the electronic properties of its surfaces to their orientation, and that may provide an experimental tool for controlling these properties. Provided rather good control on the cleaving process, various different electronic behaviors are expected on different surfaces, ranging all the way from perfect metals to insulators with varying gaps. Therefore, the WTI differs from the STI primarily in its anisotropy, and the anisotropy is not a sign of its weakness but rather of its richness². For example, the WTI constructed by 2D TI layers with weak interlayer interaction generally constrains electric transport in 1D direction to form the high anisotropic topological surface states (TSS), where only 180° backscattering is allowed distinct from the various scattering process in STI. The spin current in the anisotropic

TSS engender the ideal prohibition of backscattering and more robust nature against disorder compared to STI. We have added the related description on the advantages of WTI compared to STI in line 19 page 2 in the Introduction part as: “*Distinguished by four Z_2 invariants, the WTI differs from the STI in its surface states nature and anisotropy, which make its surface electronic properties sensitive to the surface orientation. Moreover, the spin current in WTI can be constrained in 1D direction due to the high electronic anisotropy, engendering the ideal prohibition of backscattering and more robust nature against disorder¹³⁻¹⁴.*”

Comment 2: Some additional explanations are needed concerning the calculation conditions. The use of low accuracy energy cutoff of 300 eV must be justified. What is the size of k-point meshes? I see they use the tight-binding (TB) Hamiltonian based on MLWF and the corresponding band structures should be given for the comparison with first-principles.

Reply 2: Thanks for reviewer’s careful concerns. In our calculations, the energy cutoff of 300 eV is used, and $9 \times 9 \times 1$ and $13 \times 13 \times 2$ Γ -centered k -grid meshes are adopted for structural relaxation and electronic structure calculations, respectively. The description of these parameter settings has been added in the Methods section.

To demonstrate the computational accuracy, we conducted energy convergence tests on the energy cutoff and k -grid mesh. As shown in Fig. R1, the total energy of a unit cell for $\text{Bi}_4\text{Br}_2\text{I}_2$ converge when the energy cutoff is larger than 250 eV, and the k -grid mesh along the a^* and b^* is denser than 5×5 ; while the energy convergence is achieved even the number of c^* grid points is 1. All these tests indicate that the parameter settings in our work are reasonable.

Fig. R1. Energy convergence tests on the energy cutoff (a), the number of k -grid mesh along the a^* and b^* axes (b), and along the c^* axis (c).

In addition, the tight-binding (TB) Hamiltonian based on the maximally localized Wannier functions (MLWF) is used. In Fig. R2, we compare the band structures calculated by TB model (red dashed lines) and first-principles (black solid lines). The good matching between them demonstrates that a well fitted MLWF is achieved.

Fig. R2. Comparison of bulk band structure calculated from DFT (black solid lines) and MLWF-based TB model (red dashed lines).

According to the reviewer’s suggestions, we have added Figs. R1 and R2 and related description in Supplementary Note 5.

Comment 3: The calculated band structure of $\text{Bi}_4\text{Br}_2\text{I}_2$ with and without SOC should be given in the supporting information along the suggested high-symmetry lines [Comp. Mat. Sci. 128, 140 (2017) or Comp. Mat. Sci. 49 299–312 (2010)].

Reply 3: Thanks for this nice suggestion. The calculated band structures of $\text{Bi}_4\text{Br}_2\text{I}_2$ with and without SOC along the full high-symmetry lines as the suggested references⁴⁻⁵ are plotted in Fig. R3. We have added Fig. R3 in Supplementary Note 5 and related sentence in Line 20, Page 4 in the revised manuscript as: “*The calculated results with the full high-symmetry lines could be identified in Supplementary Note 5.*”

Fig. R3. **a** and **b**, Band structures of Bi₄Br₂I₂ without **(a)** and with SOC **(b)** along the full high-symmetry lines. The red and green dots represent the weight of p_x orbital of Bi-in and Bi-ex atoms, respectively. **c**, Brillouin zone of primitive cell of Bi₄Br₂I₂, where the high-symmetry points and lines are marked by black dots and red lines, respectively.

Comment 4: I am confused concerning Fig. 4h. Do the authors mean electric gating method or charge carriers doping can tune the Fermi surface at different gap regions? If so, please give the corresponding first-principles Fermi surfaces at charge carriers doping or electric gating method.

Reply 4: Thanks for reviewer's valuable suggestions. Yes, the schematic plotted in Figure 4h means that electric gating method or charge carriers doping method can tune the Fermi surface at different gap regions. We agree with the reviewer's opinion that a detailed illustration on this issue should be provided.

First, we demonstrate that the charge carriers doping can efficiently tune the Fermi level to different binding energies. In calculations, we simulate the electron or hole doping in the system by adding or reducing additional electrons. As shown in Fig. R4, we compare the calculated surface band structures with no and 0.1 extra electrons added into the 11-layers-thick (100) slab. It can be seen that the charge doping can indeed cause the rigid shift of the Fermi level with the energy bands almost unchanged (especially for the surface state bands), and 0.1 electrons doping can induce a Fermi energy shift of almost the same energy value (0.106 eV). Hence, it is feasible to tune the Fermi surface at different gap regions simply by electric gating method or charge carriers doping. In the following we show the first-principles Fermi surfaces just by tuning the Fermi level to some certain binding energies.

Fig. R4. The (100) surface band structure without additional charges (a) and with 0.1 extra electrons (b) in the 11-layers-thick slab. The Fermi level is set to zero, as emphasized by blue dashed line.

Fig. R5. Fermi surfaces of (100) surface band with five different Fermi energies (a-e) as depicted in f. The selective binding energies of E_1 , E_2 and E_3 are the same as those in the main text. The red, brown and blue colors in a-e represent the contributions from A_1 , A_2 and B layers, respectively. The red and blue arrows depict the spin textures of the surface states.

As plotted in Fig. R5, for most Fermi energies from -0.1 eV to 0.1 eV, there are six bands with weak k_z dispersion in the whole Brillouin zone (or three bands in the half Brillouin zone of $k_y > 0$), corresponding to three pairs of helical edge states, e.g., at the binding energy of E_1 , E_4 , and E_5 . While for the Fermi energy at E_2 or E_3 , where a coupling gap is opened between two edge states, only one pair of helical edge states remains. The spin texture information is also plotted in Fig. R5, consistent with the results shown in our manuscript. Consequently, as schematically plotted in Fig. 4h, the non-degeneracy of the three channels and the interlayer

interaction-induced the energy gap provide the additional degree of freedom to control the QSH channels in selective layers in $\text{Bi}_4\text{Br}_2\text{I}_2$ by charge carriers doping or electric gating method.

Based on the reviewer's comment, we have added all these results above in Supplementary Note 9.

Comment 5: Are Fig.4e-4g the charge distribution of topological surface states at one point in the Brillouin zone for one selected binding energy? More details should be given in the manuscript. I remain puzzled by the connection between Fig.4e-4g and Fig. 4h.

Reply 5: Figs. 4e-4g are the (total) charge distributions of topological surface states at one point (or three points), which are the crossing points between three surface bands and the selected binding energy along Γ -X direction in the Brillouin zone. Such simplification (k -point decomposed charge density) is based on the result of clean and ideal surface state bands in the large bulk gap. It is also noted that we have ignored the weak k_z dispersion (*i.e.*, set $k_z = 0$) and selected a half of Brillouin zone ($k_y > 0$) to identify the spin information considering the helical nature of QSH channels.

Here, we also calculate the partial (band decomposed) charge densities with a very small energy range around the selected binding energies in the entire half Brillouin zone instead of only along Γ -X. As shown in Fig. R6, the charge distributions for Fermi energy at E_1 and E_2 are almost the same as those in Fig. 4e and 4f, respectively, demonstrating the validity of the charge distribution in Fig. 4e-4g. Since the energy window for E_3 is very small, a denser k -grid is needed, requiring expensive computational burden, so we did not obtain the result of partial charge density around E_3 for comparison with Fig. 4g. The yellow color in Fig. R6b and R6c (also in Fig. 4e-4g of manuscript) indicates the positive group velocities with an up spin for the surface states in the half Brillouin zone of $k_y > 0$; while their time-reversal partners in the half Brillouin zone of $k_y < 0$ will carry the negative group velocities with a down spin. Due to the different charge distributions at different binding energies in Fig. 4e-4g, layer-selective QSH channels can be controlled by gating or charge carrier doping, as schematically shown in Fig. 4h of the main text.

Fig. R6. Surface band structure (a) and partial (band decomposed) charge densities in the entire half Brillouin zone of $k_y > 0$ with a very small energy range (~ 1 meV) around E_1 (b) and E_2 (c).

According to the reviewer's comment, we have added the details and description of charge distribution results in Line 8, Page 7 in the revised manuscript as: "*It should be noted that Figs. 4e-g are the (total) charge distributions of topological surface states at one point (or three points) in the BZ, which are the crossing points between three surface bands and the selected binding energy along Γ -X direction. More accurate calculations of charge distributions based on a very small energy range around the selected binding energies in the entire half BZ are displayed in Figure S15 in Supplementary Note 9, which show almost the same results as in Fig. 4e and 4f, and consequently conform the validity of our results.*"

Comment 6: In the manuscript, the authors explore $\text{Bi}_4\text{Br}_2\text{I}_2$ as a WTI in the light of theory and experiments. It in current form looks like a laboratory report. The authors should highlight the findings and influence in this field.

Reply 6: Thanks for reviewer's valuable advice for highlighting the significance and influence of our work in the related field. We have added them in the discussion part as: "*Apart from the previous reported WTI stacked by the degenerate QSH layer, β - Bi_4I_4 , the $\text{Bi}_4\text{Br}_2\text{I}_2$ with triple-layer structure possesses the unique electronic properties for both of fundamental research and potential electric-device applications. The triple-times inversions at two TRIMs of BZ of bulk band structure bestow the degree of tunability on the topological properties of this system by external perturbations. For example, the strain effect has been certified to be an effective mean to realize the topological phase transition by controlling the bulk band inversion⁶. The multiple band inversions make $\text{Bi}_4\text{Br}_2\text{I}_2$ a fertile ground to realize the various novel topological phases, such as HOTI, STI, and dual topological phases, after applying the strain effect to tune the band inversions. The controllable QSH channels in thin flake of $\text{Bi}_4\text{Br}_2\text{I}_2$ offer the possibility to realize the quantized conductance with multiple values and special distributions in nanoscale size, which is unprecedented platform to semiconductor industry. Furthermore, the quasi-1D nature can enhance the electron-electron interaction strength, which could result in the helical Luttinger-liquid (LL) behavior in the edge states of monolayer $\text{Bi}_4\text{Br}_2\text{I}_2$ and the consequent coupled 2D LL behavior in the (100) surface states with weak interlayer interaction.*"

Comment 7: Besides, I would like the authors illustrate their conclusion from the physical model.

Reply 7: Thanks for reviewer's constructive suggestion. As we know, the monolayer of Bi_4X_4 or $\text{Bi}_4\text{Br}_2\text{I}_2$ (found in this work) is a QSH insulator. Here we start from the edge state Hamiltonian of QSH, and introduce

the interlayer coupling on the (100) surface to build the surface state models, which are then discussed to compare with the DFT or ARPES results.

The Hamiltonian of edge states of a QSH can be written as:

$$H_0 = \hbar v_F k_y \sigma_z, \quad (\text{R1})$$

where we have supposed the edge state channel is along the y direction, consistent with the chain direction of Bi_4X_4 or $\text{Bi}_4\text{Br}_2\text{I}_2$, and σ_z denotes the spin degree of freedom. Such helical spin-polarized bands are shown in Fig. R7a.

The bulk material is accumulated by QSH insulators with different stacking orders with weak vdW force. For $\beta\text{-Bi}_4\text{I}_4$, only a single Bi_4I_4 block is arranged along lattice c axis, engendering the formation of WTI phase. The (100) surface of $\beta\text{-Bi}_4\text{I}_4$ remains all the QSH edge states. Here and henceforth, ignoring the k_z dispersion, its surface state Hamiltonian is the same as that of QSH edge states, *i.e.*, $H_1 = H_0 = \hbar v_F k_y \sigma_z$.

For $\alpha\text{-Bi}_4\text{I}_4$ and $\alpha'\text{-Bi}_4\text{Br}_4$, there are two blocks with layer glide and/or layer rotation as the unit cell, resulting in two nondegenerate edge states. In this case, the energy gap opens at the cross points of two edges states in the reciprocal space due to the interlayer coupling. Thus, their (100) surface state Hamiltonian can be written as:

$$H_2 = \hbar v_F k_y \sigma_z \tau_0 + m \sigma_y \tau_y + \varepsilon \tau_z, \quad (\text{R2})$$

where τ denotes the layer degree of freedom, m and ε are the coupling and onsite energy difference between two edge states from two different layers respectively. In Eq. (R2), we have supposed the same Fermi velocity of these two edge states. With different Fermi velocities and asymmetric potential for two layers, the above Hamiltonian is further written as:

$$H'_2 = \begin{pmatrix} \hbar v_{F1} k_y + \varepsilon_1 & 0 & 0 & -m \\ 0 & -\hbar v_{F1} k_y + \varepsilon_1 & -m & 0 \\ 0 & -m & \hbar v_{F2} k_y + \varepsilon_2 & 0 \\ -m & 0 & 0 & -\hbar v_{F2} k_y + \varepsilon_2 \end{pmatrix}. \quad (\text{R3})$$

The corresponding band with specific values of parameters is shown in Fig. R7b. Two helical edge states with an interlayer-coupling gap are formed on the (100) surface of $\alpha\text{-Bi}_4\text{I}_4$ and $\alpha'\text{-Bi}_4\text{Br}_4$, which is consistent with the DFT and ARPES results in previous report⁷.

For $\text{Bi}_4\text{Br}_2\text{I}_2$, there are three blocks in a unit cell, resulting in three nondegenerate edge states. The energy gap opens at the cross points of every two edges states due to the interlayer coupling. Its (100) surface state Hamiltonian can be written as:

$$H_3 = \begin{pmatrix} \hbar v_{F1} k_y + \varepsilon_1 & 0 & 0 & -m_{12} & 0 & -m_{13} \\ 0 & -\hbar v_{F1} k_y + \varepsilon_1 & -m_{12} & 0 & -m_{13} & 0 \\ 0 & -m_{12} & \hbar v_{F2} k_y + \varepsilon_2 & 0 & 0 & -m_{23} \\ -m_{12} & 0 & 0 & -\hbar v_{F2} k_y + \varepsilon_2 & -m_{23} & 0 \\ 0 & -m_{13} & 0 & -m_{23} & \hbar v_{F3} k_y + \varepsilon_3 & 0 \\ -m_{13} & 0 & -m_{23} & 0 & 0 & -\hbar v_{F3} k_y + \varepsilon_3 \end{pmatrix} \quad (R4)$$

where m is the coupling between arbitrary two different layers (denoted by subscripts) and ε is the onsite potential of different layers. Its corresponding band with specific values of parameters is shown in Fig. R7c. Three helical edge states with interlayer-coupling gaps are formed on the (100) surface of $\text{Bi}_4\text{Br}_2\text{I}_2$, which is roughly consistent with our DFT calculations. In Fig. R7d, we fit our model to the DFT calculated band structure, and obtain the reasonable parameters as listed in the caption of Fig. R7, where the subscripts of 1, 2 and 3 represent the A_2 , B, and A_1 layers, respectively. Thus, our Hamiltonian of Eq. (R4) can well describe the band of surface states of $\text{Bi}_4\text{Br}_2\text{I}_2$.

Based on the reviewer's comment, we have added all these model results in Supplementary Information and the related description in the Discussion part in the revised manuscript as: "*By introducing the interlayer coupling to edge state model of a QSH (Supplementary Note 10), here we build a surface state Hamiltonian to reveal the physical process accounting for the (100) surface states of $\text{Bi}_4\text{Br}_2\text{I}_2$, which can be written as:*

$$H_S = \begin{pmatrix} \hbar v_{F1} k_y + \varepsilon_1 & 0 & 0 & -m_{12} & 0 & -m_{13} \\ 0 & -\hbar v_{F1} k_y + \varepsilon_1 & -m_{12} & 0 & -m_{13} & 0 \\ 0 & -m_{12} & \hbar v_{F2} k_y + \varepsilon_2 & 0 & 0 & -m_{23} \\ -m_{12} & 0 & 0 & -\hbar v_{F2} k_y + \varepsilon_2 & -m_{23} & 0 \\ 0 & -m_{13} & 0 & -m_{23} & \hbar v_{F3} k_y + \varepsilon_3 & 0 \\ -m_{13} & 0 & -m_{23} & 0 & 0 & -\hbar v_{F3} k_y + \varepsilon_3 \end{pmatrix} \quad (I)$$

where m is the coupling between arbitrary two different layers (denoted by subscripts and 1, 2 and 3 represent the A_2 , B, and A_1 layers, respectively), and ε and v_F are the onsite potential and Fermi velocity of different layers, respectively. Three helical edge states with interlayer-coupling gaps are formed on the (100) surface of $\text{Bi}_4\text{Br}_2\text{I}_2$, which is roughly consistent with our DFT calculations (Supplementary Note 10). We fit our model to the DFT calculated band structure to obtain the reasonable parameters, as listed in the caption of Fig. S16. All the physical parameters, m , ε and v_F , are different for three layers, confirming the non-degeneracy of three elemental layers from the view of physical model. Furthermore, our model could build a unified understanding of the formation of (100) surface states for Bi_4X_4 family of materials (Supplementary Note 10)."

Fig. R7. **a-c**, Bands calculated using the surface Hamiltonian models, i.e., Eqs. (R1-R4). The color from blue to red denotes the eigenvalues of s_z . **d**, Comparison of bands between Hamiltonian model (colored lines) and DFT calculations (black lines). The parameters are: $\hbar v_F = 0.5 \text{ eV}\cdot\text{\AA}$ in **a**; $\hbar v_{F1} = \hbar v_{F2} = 0.5 \text{ eV}\cdot\text{\AA}$, $\varepsilon_1 = -\varepsilon_2 = 0.15 \text{ eV}$, $m = 0.1 \text{ eV}$ in **b**; $\hbar v_{F1} = \hbar v_{F2} = \hbar v_{F3} = 0.5 \text{ eV}\cdot\text{\AA}$, $\varepsilon_1 = 0.08 \text{ eV}$, $\varepsilon_2 = 0.0 \text{ eV}$, $\varepsilon_3 = -0.3 \text{ eV}$, $m_{12} = 0.05 \text{ eV}$, $m_{13} = m_{23} = 0.008 \text{ eV}$ in **c**; $\hbar v_{F1} = 2.836 \text{ eV}\cdot\text{\AA}$, $\hbar v_{F2} = 2.432 \text{ eV}\cdot\text{\AA}$, $\hbar v_{F3} = 2.538 \text{ eV}\cdot\text{\AA}$, $\varepsilon_1 = 0.002394 \text{ eV}$, $\varepsilon_2 = -0.01481 \text{ eV}$, $\varepsilon_3 = -0.06453 \text{ eV}$, $m_{12} = 0.00406 \text{ eV}$, $m_{13} = 0.000186 \text{ eV}$, $m_{23} = 0.000664 \text{ eV}$ in **d**.

Reviewer 2:

This manuscript reports on a multi-technique (XRD, STM, STEM, ARPES) characterization a Bi_4X_4 -variant heterostructure based on $\text{Bi}_4\text{Br}_2\text{I}_2$ monolayers with a unique triple-layer stacking arrangement. While the ARPES of the (001) cleave-surface of the Gamma and M-bar points look very similar to the half-dozen other Bi_4I_4 and Bi_4Br_4 ARPES measurements dating back to 2015, what stands out in this work is the exceptional clarity of the ARPES spectra for the side-cleave (100) surface that reveals a cluster of three linear Dirac crossing points just below the Fermi level, which generically looks like two Dirac cones shifted in energy relative to each other.

The authors are able to reproduce the complex (100) ARPES spectra with DFT band theory, including the prediction of a third small-energy-shifted linear band dispersion, which is also experimentally inferred from the ARPES EDC line-shape with Lorentzian peak-fitting. The DFT calculations then go beyond the experimental resolution to elucidate the predicted tiny energy gappings versus degeneracies at the various linear band crossing points. DFT real-space projections of the charge densities at the different Dirac energies finds a clear localization topological surface states to specific layers of the triple-layer structure. This then leads to the logical conclusion that by tuning the Fermi-level higher or lower in energy, e.g. by gating, one could select which layer's topological edge-state contributes to the Fermi-level conduction.

The presented ARPES data are of high quality, and the auxiliary STEM and STM data are very good also. The theory presentation is clear with associated schematic figures. Hence, I can recommend publication of the current manuscript as is.

I find that another ARPES manuscript studying the same $\text{Bi}_4\text{Br}_2\text{I}_2$ material and trilayer stacking is also posted on the arXiv (Noguchi et al., arXiv:2301.07158). Again, their (001) surface results making use of laser and synchrotron energies are very similar to the other binary Bi_4X_4 ARPES studies and give the same energetics of the Gamma and M-bar bulk valence band maxima as in this study. They do not observe the detailed trilayer-splitting of topological surface state bands on the side-cleave (100) surface. This is perhaps related to their observation of both (001) and (100) surfaces being exposed after cleavage and having to use 1 micron beamspot nano-ARPES to help select which top plane is being measured. Sounds like a materials synthesis issue.

Minor: "linear" is mistyped as "liner" in a few places.

Reply: We sincerely appreciate the reviewer's highly approbatory comments about this work. We believe this work will inspire more high-quality research on tunable quantum spin Hall channels in bulk materials,

contributing to the achievement of non-dissipative and low-consumption conductive channels in the applications of nano devices. We have corrected the mentioned typo in the revised manuscript.

Reviewer 3:

Zhong et al. present structural and spectroscopic data, as well as electronic structure calculations, for the novel topological quantum material $\text{Bi}_4\text{Br}_2\text{I}_2$, built from the stacking of large-gap quantum spin Hall (QSH) Bi_4X_4 layers. The stacking of the layers in $\text{Bi}_4\text{Br}_2\text{I}_2$ is unlike that of the previously known Bi_4I_4 and Bi_4Br_4 members of the same family. The authors argue that $\text{Bi}_4\text{Br}_2\text{I}_2$ is a weak topological insulator, and that the topological edge states of the three distinct Bi_4X_4 layers in the unit cell hybridize, giving rise to a unique gap structure. This conclusion is based on high-resolution ARPES data and on first-principles calculations.

The crucial point here is the description of the surface states of the (100) surface. I find the interpretation of the ARPES results reasonable and qualitatively consistent with the experimental data. The quantitative conclusions rest on the detailed line shape analysis of the spectra of Fig. 3 (h,j). However, considering the level of accuracy required by the very small predicted gaps, a 5-peak fit provides at most circumstantial evidence. In the end, one really should trust the electronic structure calculations, and take a leap of faith (which, incidentally, is not uncommon for similar published ARPES work).

That said, the growth of high-quality crystals of a novel topological material is an achievement. Moreover, the suggestion that the Fermi level could be tuned into non-equivalent regions of the surface electronic structure, opens interesting perspectives for the control of the spin conductance of the one-dimensional topological states, and should stimulate dedicated transport measurements. Therefore, I think that the paper could be published in Nature Commun. after the authors have addressed the following points.

Comment 1: Given the importance of the electronic structure calculations, the authors should discuss how they took into account the random Br/I substitution within the layers. In particular, are they confident that the associated random potential does not smear or otherwise modify the surface gap structure?

Reply 1: Thanks for the reviewer's valuable comments. To study disordered system such as alloys and solid solutions with first-principles calculation method, approximations must be employed to take into consideration. The direct type is to build a supercell with random distribution of atoms, but such calculations generally require the use of very large supercells in order to imitate the distribution of local chemical environments, and tend to be computationally very demanding. A much simpler and computationally less expensive approach is to employ the virtual crystal approximation (VCA), in which one studies a crystal with the primitive periodicity, but composed of fictitious virtual atoms that interpolate between the behavior of the atoms in the parent compounds.

The implementation of VCA in VASP closely follows the methodology suggested by Bellaiche and Vanderbilt⁷, in which the pseudopotentials of a virtual atom are mixed from the pseudopotentials of its parent atoms according to specific weight:

$$V_{\text{ps}}(\mathbf{r}, \mathbf{r}') = \sum_{\alpha} \omega_{\alpha} V_{\text{ps}}^{\alpha}(\mathbf{r}, \mathbf{r}'), \quad (\text{R5})$$

where, V_{ps} and V_{ps}^{α} is the pseudopotentials of virtual atom and its parent atoms respectively, ω_{α} is the weight. If the pseudopotentials of this parent atoms involved in the mixing are not vastly different, this approximation can deliver satisfactory results, e.g., in $\text{Pb}(\text{Zr}_{0.5}\text{Ti}_{0.5})\text{O}_3$, GeSn alloy, $\text{Al}_x\text{In}_{1-x}\text{P}$ ⁸⁻¹¹.

In this work, we design a virtual atom at the Br/I site, whose pseudopotential is mixed from 0.5 weight of Br atoms and 0.5 weight of I atoms, to simulate the uniform and disordered distributions of Br/I found in experiments. Since Br and I are both VIIA group elements, separated by only one period in the periodic table, their pseudopotentials are likely to be similar, making the results of VCA reliable. Importantly, by employing the method of VCA, our calculated surface band structure matches well with the experimental ARPES measurements. As for the calculated energy gap of surface states, it should exist due to the allowed interlayer coupling, but there may be a small deviation in its size.

Based on the reviewer's comment, we have added some discussions above in Supplementary Note 11.

Comment 2: The description (lines 114-115) of Fig. 2(f) is poorly phrased, and as such inaccurate.

Reply 2: We have adjusted the relative description in line 4, page 5 in the revised manuscript as: “*We also observe the bottom edge of the BCB near the Fermi surface at \bar{M} point (Fig. 2e), where the approximate 0.1 eV gap to BVB is identified by the energy distribution curve (EDC) spectra in Fig. 2f.*”

Comment 3: The A1-A2-B labelling of the layers in Fig. 1(d) and 1(e) is inconsistent (twice as many labels in (e)).

Reply 3: We agree with the reviewer's comments on the inconsistent labels with the schematics of crystals structures in Figure 1(e). In order to avoid unnecessary confusion, we have replotted arrow labels of Figure 1(e) in the revised manuscript, as shown in Fig. R8.

Figure R8. Crystal structure of $\text{Bi}_4\text{Br}_2\text{I}_2$. **a**, XRD spectra of (100) plane (upper panel) and (001) plane (lower panel), respectively. The insets show the optical images of two cleaved surface with the green arrow indicating the chain direction. **b**, Large-area STM topography of cleaved (001) plane with the step height ~ 1 nm. The scale bar is 12 nm. **c**, High-resolution STM image of (001) plane with the projected schematic of atom model. The scale bar is 1 nm. **d**, HAADF-STEM results of (010) plane with alternating A_1 , A_2 , and B layers. The white rhomboid represents the triple-layer unit cell. The scale bar is 1.7 nm. **e**, (001) monolayer of Bi_4X_4 topological materials assembled by quasi-1D molecular chain as the building block along a axis by vdW force. The blue and grey balls represent Bi atoms and I/Br atoms, respectively. The schematics of the stacking mode along c axis and crystal structure of $\alpha\text{-Bi}_4\text{I}_4$, $\alpha'\text{-Bi}_4\text{Br}_4$, and $\text{Bi}_4\text{Br}_2\text{I}_2$ are shown in the orange dashed square. Each of the light red, deep red, and blue parallel arrow represents one (001) plane of Bi_4X_4 with the orientations and displacements.

Comment 4: In Supplementary Note 6, “EF moves downward” should be “bands move downward” or else “EF moves upward”.

Reply 4: Thanks for reviewer’s scrutinization of the description in supplementary materials. We have corrected the corresponding sentence in Supplementary Note 6.

Reference

1. Fu, L. & Kane, C. L. Topological insulators with inversion symmetry. *Phys. Rev. B* **76**, 045302 (2007).
2. Ringel, Z., Kraus, Y. E. & Stern, A. Strong side of weak topological insulators. *Phys. Rev. B* **86**, 045102 (2012).
3. Mong, R., Bardarson, J., Moore, J. Quantum transport and two-parameter scaling at the surface of a weak topological insulator. *Phys. Rev. Lett.* **108**, 076804 (2012).

4. Hinuma, Y., Pizzi, G., Kumagai, Y., Oba, F., Tanaka, I. Band structure diagram paths based on crystallography. *Comput. Mater. Sci.* **128**, 140-184 (2017).
5. Setyawan, W., Curtarolo, S. High-throughput electronic band structure calculations: Challenges and tools. *Comput. Mater. Sci.* **49**, 299-312 (2010).
6. Zhang, P. *et al.* Observation and control of the weak topological insulator state in ZrTe₅. *Nat. Commun.* **12**, 406 (2021).
7. Noguchi, R. *et al.* Evidence for a higher-order topological insulator in a three-dimensional material built from van der Waals stacking of bismuth-halide chains. *Nat. Mater.* **20**, 473-479 (2021).
8. Bellaiche, L. & Vanderbilt, D. Virtual crystal approximation revisited: Application to dielectric and piezoelectric properties of perovskites. *Phys. Rev. B* **61** (12), 7877-7882 (2000).
9. King-Smith, R., & Vanderbilt D. Theory of polarization of crystalline solids. *Phys. Rev. B* **47**, 1651-1654 (1993).
10. Resta, R. Macroscopic polarization in crystalline dielectrics: the geometric phase approach. *Rev. Mod. Phys.* **66**, 899-915 (1994).
11. Eckhardt, C., Hummer, K., Kresse, G. Indirect-to-direct gap transition in strained and unstrained Sn_xGe_{1-x} alloys. *Phys. Rev. B* **89**, 165201 (2014).
12. Fadila, M., Nadir, B., El-Houda, F. The elastic constants and related mechanical properties of Al_xIn_{1-x}P. *Emerg. Mater. Res.* **9**, 1060-1065 (2020).

Reviewers' Comments:

Reviewer #1:

Remarks to the Author:

I appreciate the authors' efforts to clarify the issue that I raised in my report. I recommend its publication in Nature Communications.

Reviewer #3:

Remarks to the Author:

In their rebuttal, the authors have addressed my comments. They have also provided detailed and, as far as I can tell, satisfactory replies to the remarks from the other two reviewers. In my opinion, the paper can now be published.